# Change over Time in the Mechanical Properties of Geosynthetics Used in Coastal Protection in the South-Eastern Baltic

Boris Chubarenko [1], Dmitry Domnin [1,*], Franz-Georg Simon [2], Philipp Scholz [2], Vladimir Leitsin [3], Aleksander Tovpinets [3], Konstantin Karmanov [4] and Elena Esiukova [1]

1   Shirshov Institute of Oceanology, Russian Academy of Sciences, 117997 Moscow, Russia
2   BAM Bundesanstalt für Materialforschung und -prüfung, 12205 Berlin, Germany
3   Laboratory of Fundamental and Applied Materials Science, Immanuel Kant Baltic State Federal University, 236016 Kaliningrad, Russia
4   Regional Coastal Protection Enterprise "Baltberegozaschita", 238560 Svetlogorsk, Russia
*   Correspondence: dimanisha@gmail.com

**Abstract:** The most massive design on the Baltic shore used geosynthetic materials, the landslide protection construction in Svetlogorsk (1300 m long, 90,000 m$^2$ area, South-Eastern Baltic, Kaliningrad Oblast, Russian Federation) comprises the geotextile and the erosion control geomat coating the open-air cliff slopes. Due to changes in elastic properties during long-term use in the open air, as well as due to its huge size, this structure can become a non-negligible source of microplastic pollution in the Baltic Sea. Weather conditions affected the functioning of the structure, so it was assessed that geosynthetic materials used in this outdoor (open-air) operation in coastal protection structures degraded over time. Samples taken at points with different ambient conditions (groundwater outlet; arid places; exposure to the direct sun; grass cover; under landslide) were tested on crystallinity and strain at break. Tests showed a 39–85% loss of elasticity of the polymer filaments after 3 years of use under natural conditions. Specimens exposed to sunlight are less elastic and more prone to fail, but not as much as samples taken from shaded areas in the grass and under the landslide, which were the most brittle.

**Keywords:** geosynthetics; geotextiles; properties; marine littering; contamination; coastal protection structures

## 1. Introduction

The action of wind, waves, and currents erodes the coast in the South-Eastern Baltic [1] (Figure 1). The geological structure and the deficit of free sand material at the bottom of the slope [2] make coastal protection an essential part of regional coastal management. Joint manifestation of the global trend of the rise in the world's ocean level in the South-Eastern Baltic [3] and the local trend of the increase in the water volume of the Baltic Sea because of an increase in precipitation and river runoff [4,5] increase the threat of coastal erosion [1].

In addition traditional materials [6], modern geosynthetic materials are also used to preserve the coast in various parts of the Baltic Sea and throughout the world. These materials are used for retaining walls, groins, and breakwaters [7,8]. For some regions, such as South and Southeast Asia, the inaccessibility and high cost of natural materials is a major barrier to the construction of traditional structures such as dams and breakwaters. This forces the use of geosynthetics [9]. The experience of Australia, Germany, Lithuania, Poland, the USA, South Africa, and Southeast Asia proves the effectiveness of geotextile enclosing systems for coastal protection [10–14].

In the event of damage or prolonged use, geosynthetic materials can break off and enter the marine environment [15–20]. In [21] it is indicated that on the coast of the South-Eastern Baltic (in the Kaliningrad Oblast of Russia and the Pomeranian Voivodship of Poland) there are more than 30 coastal protection and anti-landslide structures, occupying

about 20 km of the coast, and containing geosynthetic materials, which are potential sources of this types of marine and coast pollution.

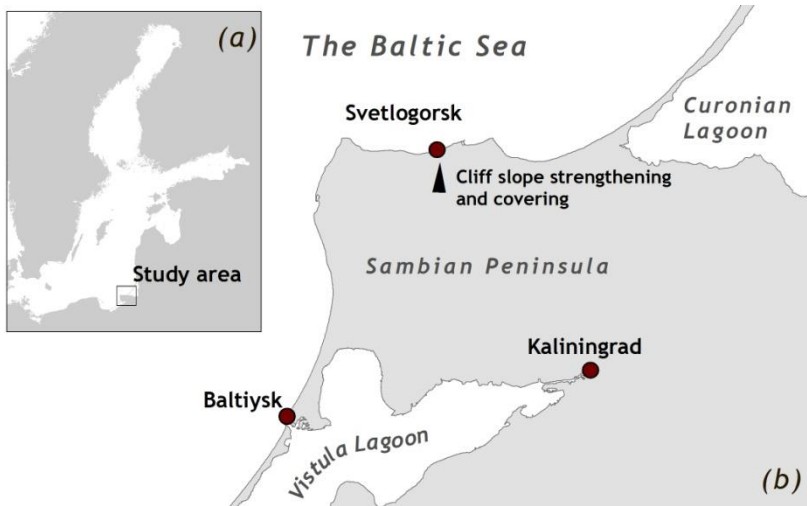

**Figure 1.** The coastal landslide protection construction, the biggest in the Baltic Sea region (**a**), is located in the City of Svetlogorsk (Kaliningrad Oblast, Russia) at the northern shore of the Sambian Peninsula (**b**).

The most common types of geosynthetics used in coastal protection structures are geo­textiles and geomats (three-dimensional fabric made of randomly arranged polypropylene monofilaments). These types of geosynthetics are contained in almost all modern structures in the South-East Baltic—on the Polish shore in 100%, and on the Russian shore in 65% of cases [21]. One of these structures attracts special attention because of its huge size. This is a complex of engineered strengthening of cliff slopes in the City of Svetlogorsk (further, Svetlogorsk) with a total area of 90,000 m$^2$ and a length of approximately 1300 m [21], where geomats are used as a continuous slope coating and geotextiles are used at the base of the slope.

This structure protects the abrasion cliff slope, which is typical for the South-East Baltic, and is common both for abrasion shores at the Russian and Polish coasts [22]. Due to the exceptionally large size and the amount of geosynthetic material used, this complex in the South-Eastern Baltic is a potential source of plastic residues and, of course, microplastic particles, which will be easily generated in the surf zone [23] if the used coating begins to degrade and parts of it enter the aquatic environment. Analysis of samples in the Baltic Sea [24] showed an increased content of microplastics in the form of threads, the main component of geosynthetic materials used in hydraulic constructions.

Coastal protection systems made of geosynthetics can be affected by metocean condi­tions (i.e., combined wind, wave, and climate conditions) as well as by the failure of the materials used [25]. The present investigation is related to the materials issue. It has been proven by various experimental studies that the service lifetime of geosynthetics is in the range of centuries [26]. The main drivers for the degradation of geosynthetic materials are thermo-oxidation and, even more effective, photo-oxidation [27].

The research questions of this work were: 1. the identification of the polymers used for the coastal protection systems and the debris found on the beach and 2. to assess what external conditions lead to the greatest change in the mechanical properties of geosynthetic materials over time, their loss of elasticity, and, accordingly, the increase in their brittleness.

The method includes laboratory analysis of samples taken at several points of the anti-landslide installation in Svetlogorsk. The sampling sites had different conditions in terms of moisture and exposure to sunlight. However, it should be noted that to preserve the integrity of the structural elements, only single test samples in the key areas of the slope were allowed by authorities to be taken.

## 2. Study Area

The glacial deposits (high, unstable coastal cliffs) of the northern shore of the Sambia Peninsula near Svetlogorsk (Figure 2a) are under constant threat of landslides. The coastal cliff of Svetlogorsk comprises alternating layers of sands, sandy loams, clayey loams, clays, and coarse-grained sediments (predominantly unconsolidated) and is easily erodible [28,29]. Coastal erosion at the toe of the slope coupled with surplus rainfall, numerous groundwater outlets, and freeze-thaw weathering are the main triggers of landslides [28]; the coastal retreat (before the cliff was fixated) was nearly the highest at the northern shore of the Sambia Peninsula. It reached 1.5 m/year on average over the long term [2].

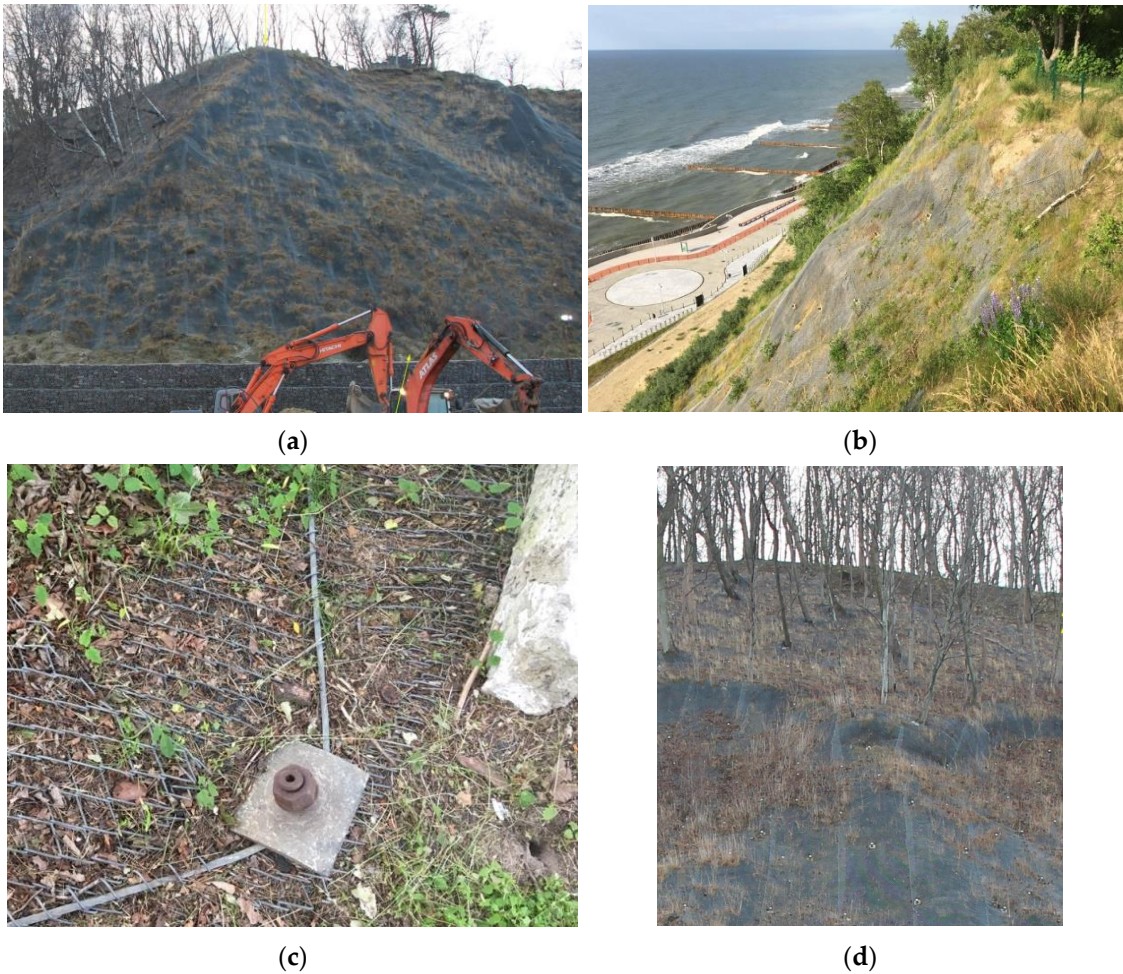

(**a**)  (**b**)

(**c**)  (**d**)

**Figure 2.** Cliff slope in Svetlogorsk during construction (**a**) and after completion (**b**) of the slide protection, which includes a geomat layer armoring of landslide slopes (**c**) and flexibly adjusted (**d**) to existing tree cover.

Erosion processes are largely related to the climatic conditions of the region and their significant changes in recent years. The Kaliningrad Oblast is located in a zone of transition from a temperate maritime to a temperate continental climate with mild, changeable winters and relatively cool summers. In the period of 1973–2016, the northern coast was characterized by an average annual air temperature of +7.8 °C (a changing trend of +0.1 °C/year), average air humidity of 81% (this remains practically unchanged), an average annual rainfall of 645 mm (+22 mm/year), an average wind speed of 3.8 m/s (–0.1 m/s per year), and maximum gusts from 21 to 40 m/s.[1]

The situation of continual cliff erosion changed with the construction in 2015–2017 of the largest coastal protection structure in the South-Eastern Baltic, the complex engineered protection of landslide slopes in Svetlogorsk (Figure 2a,b). This structure includes several



levels of gabions placed at the slope's base and the geomat layer armoring landslide slopes with steel mesh on slopes having an inclination of 1:1.5–1:2 (Figure 2c,d).

Engineered protection measures on landslide slopes in Svetlogorsk were carried out in accordance with recommendations [30,31] with the task of stabilizing the cliff and fixing the grass cover and other vegetation. The anti-landslide base is a system of anchors embedded 18 m deep into the slope. Anchors were drilled into the slope to "interlock" the layers and hold the surface layer. This was supplemented by the construction of gabions and piled retaining walls, anti-slide protection of slopes with a metal mesh, and storm net trays (Figure 3).

The geotextile (non-woven material, trademark "Dornit", widely used in coastal constructions in Russia) was used as the pad interlayers [30] between gabions used as revetments. The pieces of Dornit may be released from the construction due to direct wave impact or by improper installation (Figure 4g,h, sample Ryb01).

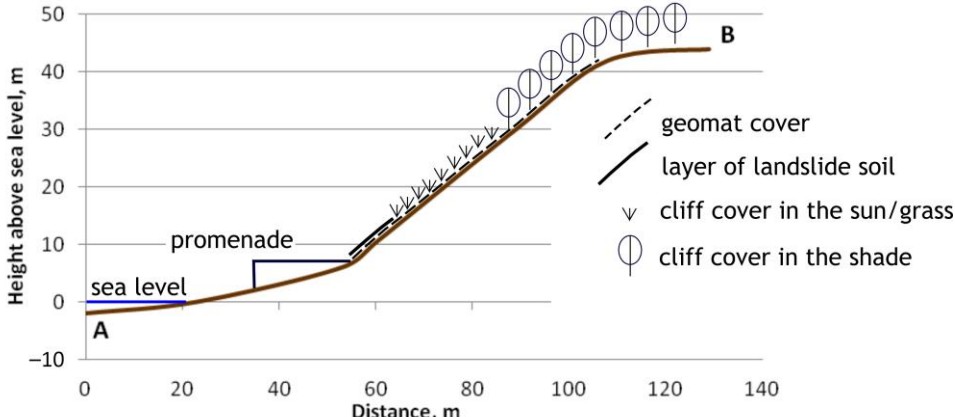

**Figure 3.** Cross profile of the slope drawn along the line AB (location see on Figure 5).

Previously, such a system was used to strengthen the slopes of the mountainous regions of southern Russia, to strengthen the slopes of engineering and transport infrastructure facilities, and during anti-karst measures. In Svetlogorsk, it was used to strengthen the coast, which is affected by both landslide and abrasion processes, unlike in previous applications.

In the years when the structure was already standing and this study was carried out (2017–2020), the average annual air temperature was higher than the average for the previous period and amounted to +9.3 °C; the average humidity was 81%; the average annual precipitation was 880 mm/year; and the average wind speed was 2.6 m/s.[2]

In general, the structure holds the slope well. The service lifetime of an erosion control geomat without grass cover was declared to be 2–3 years. The survey conducted in 2018 showed that there was minor local damage to the coastal slope that had no anti-slide protection at the upper part of the slope and for the slopes with inclinations steeper that 1:2, where the armoring was not fulfilled during the constructing phase.

The geomats under the steel mesh and geotextile at the base comprise the geosynthetic content of the structure. These geosynthetic materials are either partly exposed to direct solar radiation, or are located either in the shade, in conditions of groundwater seepage, in dry places, or hidden in the grass cover (Figures 3 and 4).

The following types of natural coverage are found throughout the structure: trees; the grass cover shaded under the trees; the grass cover without shade; local landslides without a grass cover; shrubs as undergrowth; shrubs punctuated on the slope; trees on the upper edge of the slope; trees dotted or in small groups. Some variations of the coverage are illustrated in Figure 4.

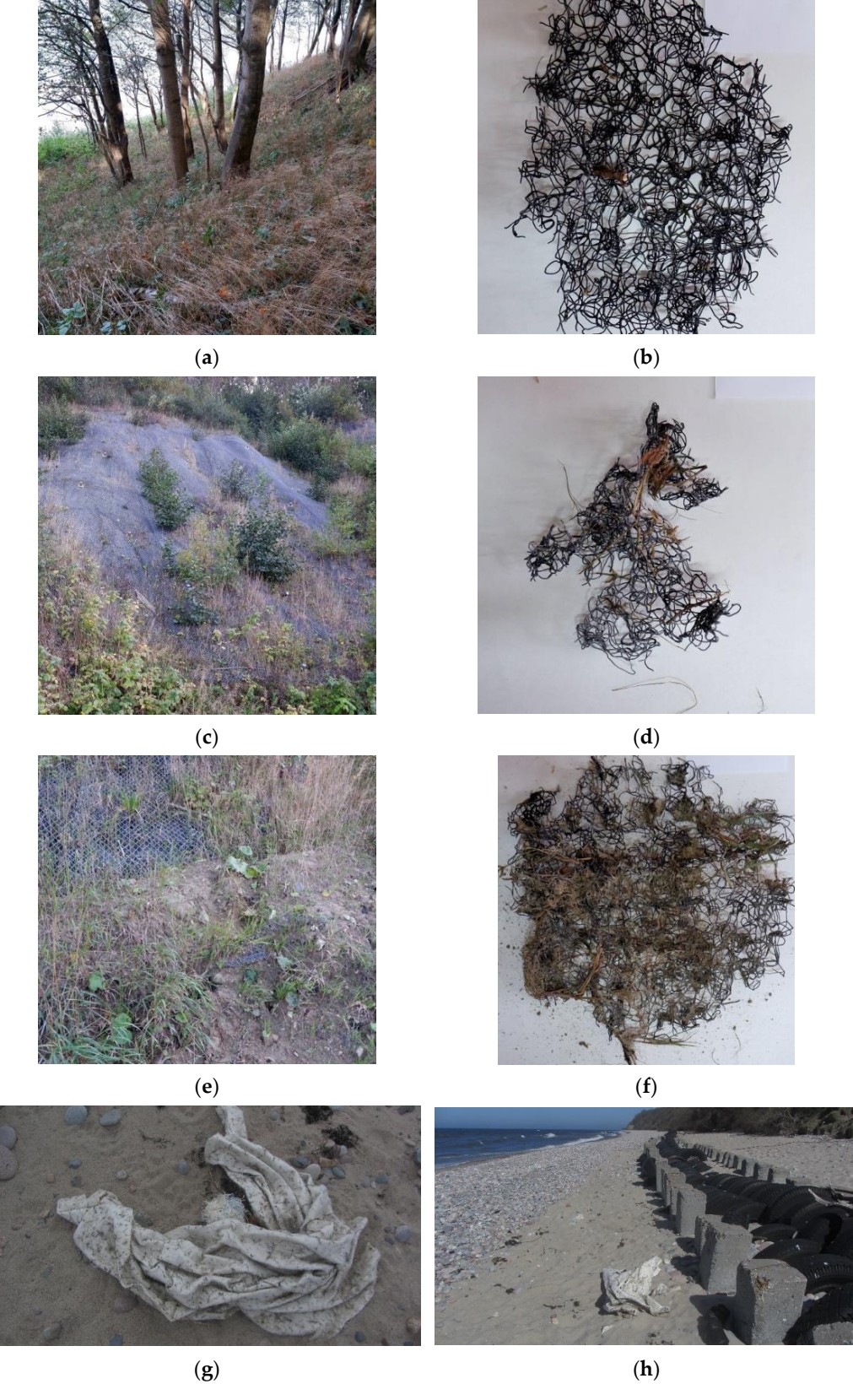

**Figure 4.** Various conditions in which the anti-slide coating is located: the upper, mostly grass-covered (**a**), and lower (**c**) (open and with small bushes) parts of the slope or covered by the landslide natural material (**e**). The samples of geomat taken in these locations (**b**,**d**,**f**). The pieces of geotextile Dornit released from the protection construction were sampled from the beach (**g**,**h**).

Several sections of the slope are completely covered with trees spaced ~2–5 m apart. The grass cover (heterogeneity was observed) on the slopes developed immediately after the facility was put into operation in 2017.

The grass cover is more pronounced in areas of the slope with trees and undergrowth, as well as in places with good illumination. It is diverse, dense, and almost completely hides the steel mesh and geomat. In strongly shaded areas of the slope, the grass cover is weakly expressed. The heterogeneity of the grass cover is associated with the degree of illumination of the slope, the unevenness of the relief, the composition of the soil, and the presence of moisture.

### 3. Materials and Methods

Ten **sampling points** (Figure 5, Table 1) were selected as test sites for different exposure conditions for the geosynthetic used in the engineered anti-slide construction. Four points were at the bottom part of the construction and five points at the upper part (heights were approximately 5 and 25 m, respectively). Due to the fact that sampling means cutting out a piece of the coating, i.e., partial destruction of the covering, permission was obtained to take the minimum possible number of samples and only at 10 points. Samples were taken carefully to avoid damage during the exhumation process. Samples of geosynthetic material were taken from the edge of the structure by cutting. This made it possible to preserve the integrity of the structure itself, and also not damage the sample. In addition, only the central part of the sample was used for laboratory analyses.

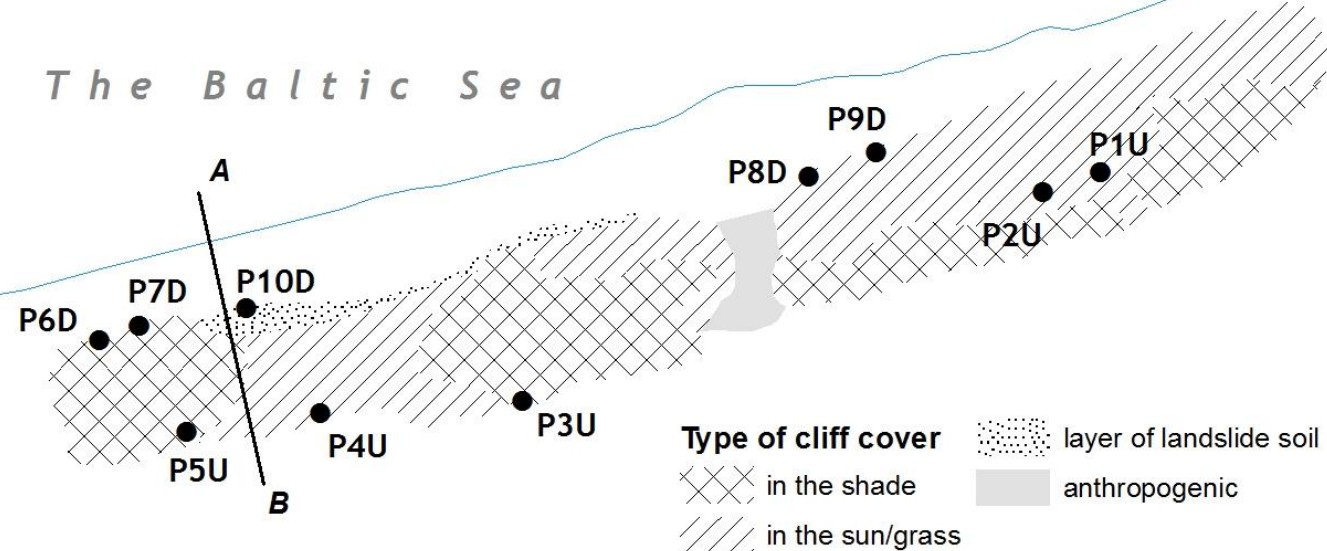

**Figure 5.** Sampling points for geomats in the complex of the structure engineered to strengthen the slopes in Svetlogorsk (of 1300 m length and up to 140 m width). The outer border of the promenade is taken as the border of the depicted coastline. The line AB shows the cross profile of the slope (Figure 3).

Damages to the materials of the cover could also be a result of the installation. However, this could not be assessed in the present investigation.

There are **different types of geosynthetic materials** (Figure 6), e.g., polypropylene (PP), polyethylene terephthalate (PET), polybutylene terephthalate (PBT), and polycyclo-hexylenedimethylene terephthalate (PCT). To identify the material composition, attenuated total reflection infrared spectroscopy (ATR-IR) was performed with a Spectrum 2000 (Perkin Elmer) device. The samples were fixed to an ATR-IR crystal (diamond) with a clamp. To increase the contact area between the sample and the crystal, the sample was melted to create a plane surface. In the first step, a background spectrum without a sample was measured using the spectrometer's control software (Spectrum, Perkin Elmer). After that,

32 spectral scans of each sample were recorded and averaged and the background was subtracted. The resulting spectra were compared with library spectra; the closest match (at least 70%) between a measured spectrum and a library spectrum was used to identify the material composition.

**Table 1.** Sampling points (Figure 5) and operating conditions, including lighting situation (exposure to sunlight or shaded).

| Point Identifier * | Latitude (° N) | Longitude (° E) | Description of the Ambient Conditions |
|---|---|---|---|
| P1U_Sun | 54.947211 | 20.161953 | The upper part of the slope in the illuminated area; the section is not covered with a steel mesh; the sample was taken from the edge, in the sun, near the tree; everything is covered with rot. |
| P1U_Sun-Grass | 54.947211 | 20.161953 | The upper part of the slope; the section is not covered with a steel mesh; the sample was taken from the edge, in the grass, under partial shade from the tree. |
| P2U_Sun-Grass | 54.947131 | 20.161564 | The upper part of the slope; the section is not covered with a steel mesh; the sample was taken at the edge of the slope in the sun. |
| P3U_Shd | 54.9463 | 20.158081 | The upper part of the slope, covered with steel mesh, in the shade. |
| P4U_Sun | 54.946244 | 20.156722 | The upper part of the slope; the section is not covered with a steel mesh; the sample was taken from the edge of the slope in the sun. |
| P5U_Shd | 54.946162 | 20.155824 | The upper part of the slope, covered with steel mesh; the sample was taken in the shade, with clay and rot. |
| P6D_Shd | 54.946516 | 20.155227 | The lower part of the slope, covered with a steel mesh; the sample was taken in the shade. |
| P7D_Shd | 54.946571 | 20.155495 | The lower part of the slope, covered with a steel mesh; the sample was taken in the shade. |
| P7D_Shd-Grass | 54.946571 | 20.155495 | The lower part of the slope, covered with a steel mesh; the sample was taken in the shade, in the grass. |
| P8D_Sun | 54.947183 | 20.159987 | The lower part of the slope, not covered with a steel mesh; the sample was taken in the sun. |
| P9D_Sun | 54.947278 | 20.160439 | The lower part of the slope, covered with a steel mesh; the sample was taken in the sun. |
| P10D_Slid | 54.946647 | 20.156215 | The lower part of the slope, not covered with a steel mesh; the sample was taken from under a layer of landslide soil. |

* The point identifier contains a serial number (from 1 to 10), the letter index corresponds to the position of the point. "U" and "D" mean the top and bottom of the slope, and the suffix describes the operating conditions (Sun—the sample was in an open sunny area, Shd—the sample was in a shady spot, Grass—the sample was covered with grass, Slid—the sample was taken from under the soil landslide layer).

To measure the **crystallinity** of the polymer material used to manufacture the geotextile and geomat samples, a DSC 7 (differential scanning calorimeter, Perkin Elmer) was used to record the melting points and cold crystallization points of the polymers. The applicable guidelines for such measurements are DIN EN ISO 11357-1,-2,-3,-6 and DIN EN ISO 11358-1.

For the measurements, approximately 5 mg of each sample were weighed and melted in closed aluminum pans. Melting comprised 3 steps: (A) Hold at 30 °C for 3 min, (B) heat up at 10 °C/min to 280 °C, (C) hold for 3 min at 280 °C. The measurement and reference cells are continuously purged with 20 mL/min nitrogen gas flow. The resulting peak areas were integrated using the Pyris software (Perkin Elmer). Samples were melted twice to obtain a clean melting curve.

**Figure 6.** Reference materials for IR database: (**a**) Polypropylene (PP), (**b**) polyethylentere phthalate (PET), (**c**) polybutylene terephthalate (PBT), and (**d**) polycyclohexylenedimethylene terephthalate (PCT).

The **tensile strength** (cN/tex) of single fibers of the geotextiles was measured in relation to the elongation (%) of individual fibers using the Vibroscope 400 and Vibrodyn 400 devices (Lenzing, Gampern, Austria). The Vibroscope 400 is an instrument for measuring the denier (linear mass density) of individual fibers in den (g per 9000 m of yarn) or tex (g per 1000 m yarn). It works by the vibration method. An electronic pulse excites the fibers to transverse vibration under strain (300 mg preload weight clamps) at a constant length. The denier is calculated from the natural frequency of the fiber and the strain weight. Once the denier is measured, the fiber is clamped in the Vibrodyn 400 with a strain weight of 300 mg; the clamping length is 20 mm. The Vibrodyn 400 is an automatic device for testing the tensile strength of single fibers. The test is based on the principle of the constant deformation speed of the specimen in the tensile direction according to DIN 51221 and 53816. The fiber is clamped in two electromagnetically operated clamps; the upper one is connected to the force-measuring device and the lower one is movable. The traction speed is 20 mm/min. A low-path electronic force transducer is used to measure the tensile strength.

The results are expressed in the form of strain at break, which value means that the break occurs when the sample is elongated by a certain percentage. For example, if a fiber with a length of 1 cm can be elongated to 1.40 cm before it breaks, the strain at break is 40%. Single fibers were pulled out from the geotextile with tweezers. This was possible also for small debris samples found on the beach. Tests were performed in triplicates.

Measurement of the geomat filaments' tensile strength was not possible because the geomat was too brittle to extract single filaments long enough for measurements with Vibroscope and Vibrodyn. The tensile-strength testing machine Instron ElectroPuls E1000 was used to conduct studies of the mechanical break (rupture test) of pieces of the three-dimensional structure (see Figure 7b) of the geomat. The value of the load was in the range of 1–100 kg, max 1000 N, as determined by the sensor. For each sample, the number of longitudinal filaments, which accounted for the tensile load, was calculated. The tests were carried out until all longitudinal filaments of the test sample were completely broken (Figure 7).

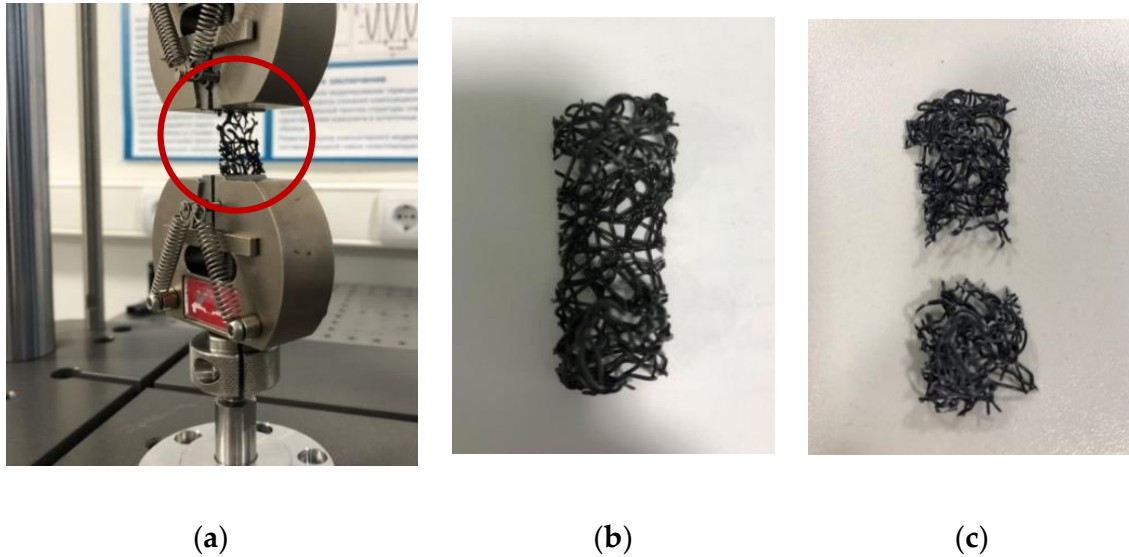

(**a**)            (**b**)            (**c**)

**Figure 7.** Rupture test of the erosion control geomat sample using (**a**) the Instron ElectroPuls E1000 and a (**b**) virgin and a (**c**) broken sample.

## 4. Results and Discussion

### 4.1. Types of Material

Two types of geosynthetic material (Figure 8) used in the structure were selected for mechanical analysis—the geotextile "Dornit ECO 300" (300 g/m$^2$ mass per unit area) and the erosion control geomat. The results of the ATR-IR analysis (Table 2) showed that geotextile is made from PET while the geomat is made from PP.

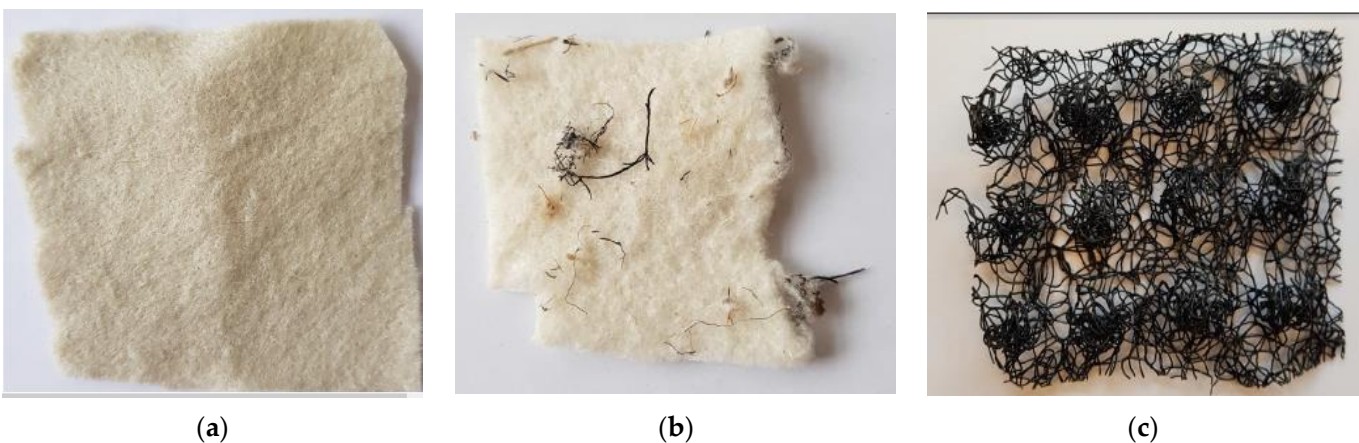

(**a**)            (**b**)            (**c**)

**Figure 8.** Analysed samples: (**a**) virgin geotextile Dornit ECO 300, (**b**) naturally aged geotextile sampled (sample Ryb01) after several months of exposure under natural conditions, and (**c**) virgin erosion control geomat.

Polypropylene is a semi-crystalline polymer, so that in solid state an amorphous phase co-exists with the crystalline phase. The degree of a polymer's crystallinity affects its physical and mechanical properties, including its tensile strength. Typical crystallinity for PP is 30–60% [32]. The measured trend of the samples' crystallinity [33,34] was proved, as aged polymer materials show greater crystallinity than virgin materials (Table 2).

**Table 2.** Identification of polymers according to IR database (probability of match) and results of the crystallinity analysis of two samples of erosion control geomat and two samples of geotextile.

| Sample | Material, Probability of Match | Crystallinity | Strain at Break |
| --- | --- | --- | --- |
| Geotextile (Dornit ECO 300, virgin), Figure 8a | 74.1% PET | 29.3% | 42.2 ± 9.8% |
| Geotextile (Ryb01, aged Dornit), Figure 8b | 71.9% PET | 39.6% | 15.5 ± 0.8% |
| **Sample** | **Material, Probability of Match** | **Crystallinity** | **Strain at Break** |
| Geomat (virgin), Figure 8c | 98.0% PP | 35.3% | See Table 3 |
| Geomat (aged) | 98.0% PP | 39.1% | See Table 3 |

**Table 3.** Filament break test results.

| Sample | Max. Load (N) | Number of Longitudinal Filaments (pcs.) | Avg. Max. Load Per Filament (N) | Maximum Deformation until Failure (mm) | Loss of Elasticity |
| --- | --- | --- | --- | --- | --- |
| Reference sample | 18 ± 0.5 | 6 | 3.0 ± 0.5 | 29 ± 0.5 | 0% (reference) |
| P7D_Shd-Grass, in the shade, covered with grass | 30 ± 0.5 | 9 | 3.3 ± 0.5 | 8 ± 0.5 | 72% |
| P7D_Shd, in the shade | 16 ± 0.5 | 10 | 1.6 ± 0.5 | 9 ± 0.5 | 69% |
| P8D_Sun, in the sun | 26 ± 0.5 | 9 | 2.9 ± 0.5 | 16 ± 0.5 | 45% |
| P10D_Slid, under the soil landslide layer | 18 ± 0.5 | 6 | 3.0 ± 0.5 | 11 ± 0.5 | 62% |

### 4.2. Strain at Break Analyses

The mechanical properties of the sample Ryb01 were investigated using individual fibers pulled out from the sample. The strain at break dropped significantly from more than 40% for the virgin sample to around 155 for the exposed samples from the beach.

The geomat samples taken in the conditions of the shadow, grass cover, open sun expose, under the slided ground were examined with the rapture test.

The reference sample (Figure 9a) and sample P7D_Shd taken in the shade (Figure 9b) showed behavior similar to each other—the load reaches the limit value. After that, 2–3 filaments are torn, and then the sample withstands approximately the same load (50–70% of the maximum), but the filaments are torn one by one. This means that the load is not borne by the entire set of filaments, but by one or a few them. Therefore, when one filament breaks, the situation changes only slightly—the next filament takes on the entire load.

Although samples P7D_Shd-Grass (Figure 9c) and P8D_Sun (Figure 9d) were under different field conditions (P7D_Shd-Grass—in the shade, covered by grass; P8D_Sun—in the sun), they demonstrate the expected behavior for samples with a set of filaments loaded approximately evenly: the load, having reached its limit, then dropped to virtually zero through a series of filament breaks.

Sample P10D_Slid taken from under the landslide layer (Figure 9e) demonstrated behavior similar to the reference sample. The load was not perceived by the entire set of filaments, but most likely by one or two. According to the diagram, each next filament withstood a greater load than the previous one. The results obtained showed that the values of the maximum load per filament for samples P7D_Shd-Grass, P_Sun, and P10D_Slid were in a range similar to those of the reference sample. As for the P7D_Shd sample, the maximum load per filament turned out to be roughly 50% of that of the reference sample.

This indicates the possible presence of significant micro-damage to the filaments in this sample.

In terms of deformation, the reference sample was found to be the most elastic. The most brittle samples were P7D_Shd-Grass (in the shade, covered with grass) with a loss of 85% and P10D_Slid (under a layer of soil landslide) with a loss of 70% of the elasticity of the reference. Samples P7D_Shd (in the shade) and P8D_Sun (in the sun) also lost elastic properties (39% and 50%, respectively) (Table 3).

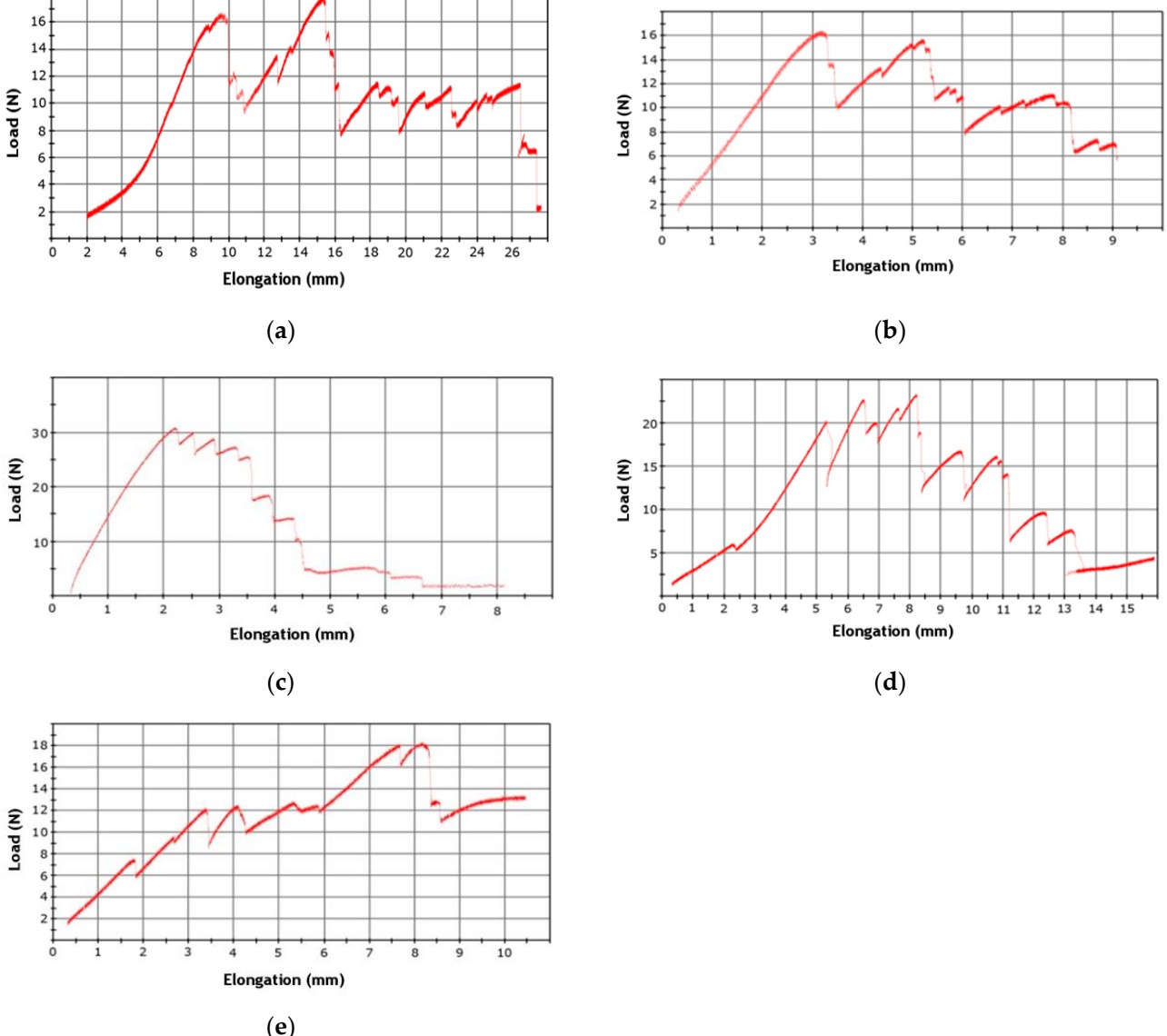

**Figure 9.** Load-elongation diagrams for (**a**) the reference sample; (**b**) sample P7D_Shd located in the shade; (**c**) sample P7D_Shd-Grass covered by grass in the shade; and (**d**) sample P8D_Sun located in the sun; and (**e**) sample P10D_Slid taken from under the landslide layer. Sharp increases in load are associated with successive filament breaks.

The presence/absence of micro-/macro-damage in the geomat and its heterogeneities (inclusions differing in X-ray density in the degree of greyness from the main material of the sample) were investigated [35]. Filament strength correlated with the observed bulk density of micro-damage—the lower the density of observed micro-damage, the higher the maximum load per filament.

The hypothesis that the material placed in the sun will lose more elasticity than material in the shade was not supported by the experimental results described above.

The method applied to assess the elasticity of the geomat samples is not a standardized procedure. However, it was demonstrated that the measured values of the unaged reference yielded the highest values for the maximum deformation before failure; the results for the aged samples were considerably lower. The reason why samples exposed to sunlight showed better results than samples from shaded areas could not be clarified.

## 5. Conclusive Remarks

The landslide protection construction in Svetlogorsk (1300 m length, 90,000 m$^2$ area, south-eastern Baltic, Kaliningrad Oblast, Russian Federation) comprises geotextile and erosion control geomat coating of the open-air cliff slopes. Due to changes in elastic properties during long-term use in the open air, as well as due to its huge size, this structure can become a non-negligible source of microplastic pollution in the Baltic Sea.

Weather conditions affected the functioning of the structure. It was assessed that geosynthetic materials used in these outdoor (open-air) coastal protection structures degraded in time. After 3 years of use under natural conditions, polymer filaments in the erosion control geomat lost 39–85% of their elasticity.

It was expected that the main factor in the degradation of geosynthetic material (geotextile or erosion control geomat) would be photo-oxidative conditions over a longer period of time. Specimens exposed to sunlight lose elasticity and are more prone to fail due to brittleness, but not as much as specimens from the shade. The present study did not confirm the expected greater loss of elasticity in the material exposed to sunlight than in the material in shaded areas. The samples taken from shaded areas in the grass and under the landslide were the most brittle. The results obtained could not reveal the reason for this behavior, which should be studied in greater detail. Further, due to the fact that geosynthetics may degrade in contact with atmospheric oxygen and sunlight, it is disputable whether such installations as in Svetlogorsk should be better constructed with biodegradable material [36].

The engineering idea for the structures in Svetlogorsk was to use the steel mesh as a power core for the anti-landslide protection, and the geomat layer as the basis for the development of ground cover vegetation, which will develop in 3–5 years. After the appearance of ground cover vegetation, the geomat can even collapse, since it has already served. In contrast to this purely utilitarian approach, we are interested in the possible negative impact on the environment due to the blowing and washing out of small fragments of the geomat, which is destroyed day by day under the influence of natural factors.

The fact that these types of geosynthetics are already polluting the marine environment and the beach has been confirmed by studies [37]. The release of geosynthetic fibers may be related to the degradation of coastal protection structures and construction activities at the shore. Only during the periods of field expeditions 2018–2020 more than 50 m$^2$ of geotextiles and numerous geomat fragments were found on the Russian, Polish and Lithuanian shores. However, geosynthetic fragments found on the beach are also the result of improper waste management [20].

The data obtained represent the first-ever estimate for the huge landslide protection construction in the south-eastern Baltic coastline and can be used in the future as a reference to assess the degree of degradation of geosynthetic materials here or for comparison with other shores. Researchers, beach managers, and practitioners can use these data also for the improvement of construction, installation, and maintenance of coastal protection systems using geosynthetics. The data may be useful as reference data on the virgin properties of the debris of geosynthetic materials found in nature.

The landslide protection construction itself is still in a good state from the engineering point of view: the structure fulfills the task of stabilizing the slope, ground cover vegetation develops on top of the synthetic cover, and mechanical destruction of the cover is currently lacking. There is no need to do any remedial work just now. The results help to identify the locations of future possible damages in the geomat cover to optimize the damage monitoring efforts.

**Author Contributions:** Conceptualization, B.C.; Methodology, B.C., D.D., F.-G.S. and V.L.; Formal analysis and investigation: sample acquisition, selection, description and archiving, D.D., K.K. and E.E.; crystallinity and tensile strength analyses on Vibroscope 400 and Vibrodyn 400 devices, P.S. and F.-G.S.; tensile strength analyses on Instron ElectroPuls E1000, A.T. and V.L.; Writing—original draft preparation, D.D. and B.C.; Writing—review and editing, B.C. and F.-G.S.; Figure preparation, D.D., P.S. and A.T.; Supervision and funding acquisition: F.-G.S. and B.C. All authors have read and agreed to the published version of the manuscript.

**Funding:** The data collection and laboratory work were supported within the ERANET-Rus project EI-GEO, and its national grants RFBR 18-55-76002 ERA_a and BMBF 01DJ18005. The data analysis was organized within the theme FMWE-2021-0012 of the State Assignment of the Shirshov Institute of Oceanology of the Russian Academy of Sciences.

**Conflicts of Interest:** The authors declare that they have no known competing financial interests or personal relationships, which have or could be perceived to have influenced the work reported in this article.

## Notes

[1]   According to the server http://www.pogodaiklimat.ru/.

[2]   According to the server http://www.pogodaiklimat.ru/.

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
