# Peer review of "Change over Time in the Mechanical Properties of Geosynthetics Used in Coastal Protection in the South-Eastern Baltic"

_jmse, doi:10.3390/jmse11010113_

Round 1

Reviewer 1 Report

The paper deals with rapid weathering of geotextile used for the landslide and erosion protection of the cliff slopes at Baltic sea shore. A case example is developed for cliff slopes in Svetlogorsk (total area of 90,000 and 1,3km length). The research work examines external causes which lead to the loss of elasticity of geosynthetic materials over time for the variety of the boundary conditions that can be found on the protected slopes. Materials and methodologies of the testing materials are clearly explained and results are interpreted with required detail. However, in terms of technical content there is a lack of insight of what measured results mean for the stability of the protected slopes. What are the consequences of these findings in the long term? Would it be necessary to do remedial works? If yes, how this findings help in finding the appropriate solutions?

Although the paper presents some novel and unexpected results there is no effort made to explain the difference and relate them to the solution of the established problem. Due to this, in my opinion, the paper lacks in terms of scientific technical content as it simply declare the results of the testing. Consequently, it is difficult to assess the importance of this work to the readers of Journal of Marine Science and Engineering. I suggest to the authors to address the issue of deficiency in the content of their scientific work and resubmit the improved paper.

In the continuation are given comments that can help authors for the future work on the paper.

1.    The abstract does not give appropriate initial information about the topic and the methodology of scientific research. Instead it contains the final results of the research which should have been presented in conclusions.

2.    The use of English language needs to be improved at certain sections. The authors are advised to engage a native speaker or use English editing services for this purpose.

Reviewer 2 Report

This work aims to evaluate the degradation over time of two geosynthetics in a real-world application. The topic of the work is important and could be a contribution to increase knowledge about the long-term behaviour of these materials. However, and unfortunately, the work should have been better developed and presented. In its current state, it does not meet the minimum requirements to be published in a scientific journal. More specific comments can be found below:

1. The introduction should include a state of the art on the durability of geosynthetics, mainly focusing in coastal protection applications. The degradation agents of these materials should have been properly identified and their effect on geosynthetics discussed.

2. The discussion/statements about the contaminating potential of geosynthetics should have been done with more care. This subject in not consensual, and there is even some controversy around this situation. In addition, this work does not allow drawing conclusions about geosynthetic-induced contamination of marine ecosystems.

3. The experimental description is sometimes confusing and important information is missing.

4. How were the samples collected? How do the authors ensure that there was no damage induced to geosynthetics during the exhumation process?

5. Degradation is evaluated by comparing the properties of exposed samples with the properties of damaged samples. Do the authors ensure that the installation process did not damage the geosynthetics? Was this controlled?

6. The description of the sampling points is not clear. The text mentions 9 sampling points, Figure 5 and Table shows 10 points. Figure 10 also shows different conditions for the same point.

7. The experimental work to identify the polymers of the geosynthetics does not contribute to the objectives and theme of the work. It is even strange that the authors did not know the basic characteristics of the materials they were studying.

8. How were the geotextile fibres obtained for the tensile tests? How do the authors ensure that removing the fibres from the nonwoven structure did not damage them?

9. Why did the authors not use standard methods (e.g. ISO 10319) for tensile tests? The choice for testing single fibres is not understandable.

10. How many replicates were tested for each sample? Information about determination errors (e.g. confidence intervals) are missing.

11. The tensile strength of geotextiles is typically expressed in kN/m, not in %.

12. Tensile strength and strain at break are two different tensile properties.

13. The description of the geomat tensile tests is not clear. What was the reason for not following the universally accepted tensile tests in the field of geosynthetics?

14. The “Results and Discussion” section should be better organised. The presentation of results should be clearer. The discussion is very underdeveloped.

15. The main characteristics (e.g. mass per unit area, thickness, tensile properties according to standard methods) of the geosynthetics under study should be presented.

16. Table 3 – Why were not all samples tested? What was the criterion for choosing only these samples? The conditions of ageing are not clear.

17. Section 3.2 – It is not clear if these results are related to the geotextile or the geomat. The tensile results for all samples should be presented, with due errors (e.g. standard deviations or confidence intervals).

18. Figure 9 – Why load and elongation have negative values?

19. From what can be seen, the state of degradation of the geosynthetics was not properly evaluated. The tensile properties of the intact samples should have been properly compared with the properties of the exposed samples.

20. The authors conclude that the specimens exposed to sunlight are prone to tearing. However, no tearing tests (e.g. ASTM D4533) were performed in this work.

21. Overall, there are several conclusions that are not adequately supported by the experimental results. As mentioned, the presentation and discussion of the results is also below expectations.

As a final note, the article does not have numbered lines, making it difficult to review.

Round 2

Reviewer 1 Report

In my opinion the authors adequately replied on the questions and introduced the necessary changes.

The authors need to specify in technical terms what does it mean that ¨landslide protection construction is still in good shape¨. They need to clarify if ˝good shape˝ is related to stability or erosion protection or some other parameters that should be monitored. If the continuous monitoring is the only current measure what would be the relevance of the findings for the monitoring programme?  

Reviewer 2 Report

The revised version of the article is not very different from the original version. Taking into account the outcome of the previous review, it was expected that the authors would have made more significant changes to the article. Most issues raised in the previous review were not adequately answered. Therefore, the article still does not have the minimum requirements to be published in a scientific journal.

The main problem of the work is the experimental plan, which should have been better developed. The authors should also have controlled all aspects of the work (from installation to characterisation of geosynthetics), which apparently did not happen. Such an investigation requires:

1. Knowing the geosynthetics and their properties. The need to identify the polymers (which adds nothing to the article) only shows lack of control on the experimental design. The goal “the identification of the polymers used for coastal protection systems…” is of no interest, as this is widely known. Furthermore, testing only 2 geosynthetics would never achieve this goal. In the conclusion, it is mentioned that “the data… can be used in future as a reference to assess the degree of degradation of geosynthetic materials here or for comparison with other shores”. How can this be useful if the properties of the tested materials are unknown? Were the tested geosynthetics properly protected from oxidation?

2. Definition of proper characterisation methods. The choice for testing single fibres removed from the nonwoven structure is not understandable. How do the authors ensure that removing a fibre (typically with a diameter of about 30 to 50 microns) from a nonwoven structure does not damage the fibre? How do the authors ensure that the changes they are observing in the tensile behaviour of the fibres are not due to damage occurring in the removal process? A standard test, such as ISO 10319, should have been used to determine the tensile behaviour of the geosynthetics.

3. The damage occurring during the installation of the geosynthetics should have been monitored. How do the authors ensure that the changes they are observing in the tensile behaviour of the geosynthetics are not due (totally or partially) to installation damage?

Other comments include:

- Introduction should have been better developed. It is strange that in an article on the durability of geosynthetics the authors refer that “it is out of scope of the manuscript to discuss this issue in detail”.

- The experimental description remains unclear and incomplete. E.g. the description of the geomats tensile tests is not clear. Which test method was used?

- Tensile strength and strain at break are two different properties (e.g. Table 2 continues not to distinguish these properties).

- The discussion continues very underdeveloped.

- The reason presented by the authors to not present thickness values is not understandable. There are methods to determine this parameter (e.g. EN ISO 9863-1). As mentioned before, a full characterisation of the geosynthetics should have been included in the article.

- The authors did not explain why they did not test all samples (comment 16 in the previous review).

- Load and elongation should not have negative values. The axis should have been corrected in order to represent real values.

- The changes made in the conclusions should have been better performed. It is not clear if the authors are taking conclusions from their results or reviewing literature. Some conclusions are not adequately supported by the results.

- The authors suggest that the structure should be better constructed with biodegradable material. How do the authors ensure that biodegradable materials will have a long-term behaviour compatible with the needs of the structure? 

Round 3

Reviewer 2 Report

Globally, the comments/suggestions are the same as those made in the previous reviews.